# Hybrid Attribution Priors for Explainable and Robust Model Training

## Abstract

Small language models (SLMs) are widely used in tasks that require low latency and lightweight deployment, especially classification. With the growing emphasis on interpretability and robustness, explanation-guided learning offers an effective framework by incorporating attribution-based supervision during training. However, how to derive general and reliable attribution priors remains an open challenge. Upon analyzing representative attribution methods in classification tasks, we find that while these methods reliably highlight class-relevant tokens, they tend to focus on common keywords shared by semantically similar classes. Since these classes are already prone to confusion under standard training, the attributions fail to provide sufficient discriminative cues, limiting their ability to enhance model differentiation. To address this challenge, we introduce **C**lass-Aware **A**ttribution **P**rior (**CAP**), a novel attribution prior extraction framework designed to guide language models in capturing fine-grained class distinctions, thus producing more salient and discriminative attribution priors. Building on this, we propose **CAP**$_{Hybrid}$, which integrates priors from CAP and existing attribution techniques to form a more comprehensive and balanced supervisory signal. By aligning the model's self-attribution with these enriched priors, our approach encourages the model to capture diverse decision-relevant features. Extensive experiments across full-data, few-shot, and adversarial settings demonstrate that our method consistently enhances both model interpretability and robustness.

## 1 Introduction

Small language models (SLMs), such as Bert (Devlin et al., 2019) and Roberta (Liu et al., 2019b), have become indispensable, particularly for classification scenarios, due to their efficient inference capabilities in resource-constrained environments and the ease of deployment on devices. SLMs power a wide range of essential applications, such as intent detection in dialogue systems (Zhang et al., 2023), sentiment analysis in social media (Wankhade et al., 2022), and topic categorization on content platforms (Li et al., 2022b). In recent years, the performance of SLMs on standard classification benchmarks has improved considerably (Wang et al., 2019), achieving competitive accuracy with significantly smaller model sizes. The focus has gradually shifted to improving the interpretability and robustness of SLMs so that they can meet the real-world demand for transparent model decision-making mechanisms (Taha et al., 2024). Some researches focus on explanation-guided learning (Liu & Avci, 2019), in which models are trained not only to predict correct labels but also to align attribution priors to identify the most relevant input components for a given prediction, thus enhancing interpretability and potentially improving robustness.

Current methods for obtaining attribution priors fall into two main categories. The first derives priors from the trained model itself or from larger, pre-trained variants of the same architecture using techniques such as attention attribution or integrated gradients (Wu et al., 2023). These self-derived priors reflect the model own insights into the data and knowledge, but can be biased, especially in small models with limited capacity. The second category involves human-annotated priors that encode linguistic or domain-specific knowledge (Jayaram & Allaway, 2021), offering greater transparency and generalizability. However, these priors are expensive to construct and lack scalability. So there raises a natural question: *How to obtain attribution priors that are both reliable and scalable, which capturing both model-internal reasoning and external linguistic knowledge?* To explore this question, we try to solve the following three problems:

Figure 1: Training model with attribution produced by different methods for the same case.

**Q1:** Do existing attribution methods exhibit systematic limitations in SLM-based text classification?

**Q2:** Large language models inherently encode rich prior knowledge and possess strong language modeling capabilities. But how can we effectively extract high-quality attribution priors from them and integrate these priors with existing methods to enhance explanation-guided learning?

**Q3:** Will the refined attribution prior consistently achieve good interpretability and robust supervision across tasks in a variety of settings?

Specifically, we first analyze the characteristics of the existing typical attribution methods on classification tasks. For the traditional self-attribution method, such as integrated gradients, they inevitably inherit biases from the training data and cannot introduce novel and complementary knowledge. For the typical LIME and SHAP methods that can extract attribution scores from LLMs to guide SLMs, we observe two key phenomena: homogenization and class confusion. First, we find that both methods produce highly correlated attribution scores and consistently highlighting tokens with high affinity to the target label. Second, the class-level analysis of the top LIME-ranked words reveals that certain classes share many frequent keywords, showing clear word overlap among various classes. Importantly, these confused classes correspond to those most often misclassified by the vanilla-trained target model. Although LIME captures class-relevant cues, it fails to emphasize discriminative features that distinguish these ambiguous classes, thus directly using such priors to supervise training may amplify errors in confused classes. As shown in Figure 1, if the prior only captures the keywords *set* and *alarm* in the set_alarm category, it may lead to more serious misjudgments for close categories with similar keywords, such as the example case from query_alarm. Ideally, the attribution would fully comprehend the semantics, and assign a high attribution score to the keyword *check* for this case.

To solve these issues, we introduce Class-Aware Attrbution Prior (CAP), a novel attribution prior extraction method that enriches the input text with task instructions and label space. By prompting LLMs in this way, CAP captures class distributions and generates more salient and discriminative attribution priors. Motivated by the complementarity among different attribution methods, we then fuse attribution priors from CAP and existing methods creating a more refined and reliable prior signal. By subsequently aligning the model's self-attributions with this rich prior knowledge during training, the SLM learns more reliable and general reasoning patterns. We evaluate our method across three diverse datasets on full-data training, few-shot learning, and adversarial settings. Extensive results demonstrate that the refined attribution prior consistently improves classification accuracy while significantly enhancing interpretability and robustness to adversarial perturbations. Our main contributions are summarized as follows:

- Through analysis of attribution scores from existing methods on text classification tasks, we identify two key limitations: homogenization and class confusion, This will lead to misclassification of semantically similar categories that share overlapping keywords under the attribution guided learning paradigm.

- We propose a class-aware attribution prior method that extracts salient and discriminative priors, along with a hybrid variant that captures more comprehensive attribution signals.

- We evaluate the effectiveness of our method across three experimental settings, focusing on accuracy, interpretability, and robustness. In addition to quantitative results, we also conduct qualitative analyses to further validate the impact of our method.

## 2 RELATED WORK

### 2.1 MODEL ROBUSTNESS AND INTERPRETABILITY

Robustness and interpretability are critical components of any responsible and trustworthy models, where their black-box nature makes them vulnerable to adversarial attacks (Jyoti et al., 2022), especially in safety-critical or risk-sensitive domains (Tjoa & Guan, 2021; Kaur et al., 2023). Several studies have investigated adversarial attacks and defenses to bolster model robustness (Zhang et al., 2021; Hendrycks et al., 2020). Likewise, recent efforts have focused on enhancing the stability of explanation techniques (Wang et al., 2020; Li et al., 2023). Researchers have examined the connections between robustness and interpretability, highlighting the elicitation of more reliable explanations as a key pathway to enhance interpretative robustness (Li et al., 2022a). For interpretability, in recent years, many explanation techniques have been developed to better understand the decision making process of deep neural networks (Bach et al., 2015; Ribeiro et al., 2016; Selvaraju et al., 2017; Lyu et al., 2024). Integrated gradients (Sundararajan et al., 2017) addresses input-sensitivity by estimating each feature contribution along the straight-line path from a baseline input to the actual input. LIME (Ribeiro et al., 2016) generates perturbed samples by masking different subsets of features around the target instance, computes the predictions on these samples, and then fits a locally weighted surrogate model to attribute importance. SHAP (Lundberg & Lee, 2017) provides a unified interpretation framework that encompasses six prior methods, assigning each feature an importance score for the given prediction.

### 2.2 ATTRIBUTION-DRIVEN LEARNING

Recent studies have shown that integrating explanation signals can enhance model performance (Gao et al., 2022a;b; Rao et al., 2023). (Ross et al., 2017) regularize models by penalizing input gradients to match expert-defined attribution maps. $e^2$KD (Parchami-Araghi et al., 2024) improves distillation faithfulness by aligning GradCAM explanations (Selvaraju et al., 2017) or B-cos model attributions (Böhle et al., 2022) between teachers and students. In natural language processing domains, several studies leverage human-annotated rationales as attribution priors. DBLP:conf/emnlp/JayaramA21 crowdsource word importance annotations on a small subset to construct attribution priors and apply them into mean attention weights attribution methods for detecting opinions and beliefs from text. (Liu & Avci, 2019) collect known toxic words and identity terms, and achieves better performance in text classification by setting an attribution target of 1 on toxic words and 0 for identity terms. DBLP:journals/corr/abs-1908-06870 propose to enhance faithful explanations of model attention for fine-grained sentiment analysis by aligning the model attention with human rationales. AD-KD (Wu et al., 2023) leverages the multi-view attribution distillation for model compression. DBLP:conf/naacl/PimparkhedeB25 propose to use main predicate in the text utterances along with the arguments of the main predicate to serve as explanation signals for intent classification. Although effective, these methods depend on costly human annotations or require parameter updates, limiting their scalability. In this paper, we propose a novel approach that automatically extracts comprehensive attribution priors from LLMs without any human effort or model fine-tuning.

## 3 ANALYSIS OF ATTRIBUTION PRIORS

To answer the **Q1**, we analyze the characteristics of existing attribution methods in this section.

### 3.0.1 INTEGRATED GRADIENT

As a typical white-box attribution method, Integrated Gradients (IG) computes token importance based on the model's internal gradient information. Following (Wu et al., 2023), a common approach in attribution-prior guided training for small language models (SLMs) is to first train a model on the dataset, then extract attribution scores from this trained model over the training data, and incorporate these scores as priors during subsequent training. This method's advantage lies in the strong alignment between the attribution priors and the model's own decision process, enabling the model to reinforce its learned reasoning patterns. However, this approach also has limitations: since the attribution depends on the initially trained model, it is susceptible to biases present in that model; moreover, IG captures only local gradient-based information and may not effectively lever-

age semantic label information or external knowledge. This limitation is particularly pronounced in few-shot or fine-grained settings, where distinguishing between semantically similar classes requires richer and more discriminative attribution signals.

### 3.0.2 LIME AND SHAP

We investigate two representative black-box attribution methods, LIME and SHAP, which are capable of extracting rationale priors from large language models' predictions. These methods provide additional explanatory information from large models that can be used to guide the training of smaller models. Firstly, we take text as the input and the corresponding label as the target output to automatically extract priors by estimating the impact of each input token on the model target output, thus yielding the correpsonding attribution scores. This leads to a question: what characteristics define high-quality attribution priors? To explore this, we analyze the attribution scores generated by LIME and SHAP on the Banking77 dataset and observe two key phenomena: homogenization and class confusion. First, both methods highlight words with high affinity to the target label. We also calculate the Pearson correlation coefficient between two scores, and find that it reaches 0.8, suggesting that the two methods focus on the similar subset of words. Second, we examine class-level behavior by identifying the words with higher LIME attribution scores for each class. By calculating the lexical overlap across classes, we observe that while many classes have distinct keywords, a subset exhibits substantial overlap in their attributed terms, called confused classes. Directly using such priors to supervise training may amplify errors in these categories, as they encourage the model to focus on overlapping words.

We further analyze the relationship between the misclassified classes from models trained solely with classification loss and the confused classes. Fgiure 2 show the relatinonshaip between misclassified samples under standard training and confused classes of LIME. The horizontal axis shows the top 15 classes sorted by the number of misclassifications, with higher values indicating more frequent errors. The vertical axis represents the degree of keyword overlap between class pairs based on LIME attributions, where higher values indicate greater overlap. As shown in Fgiure

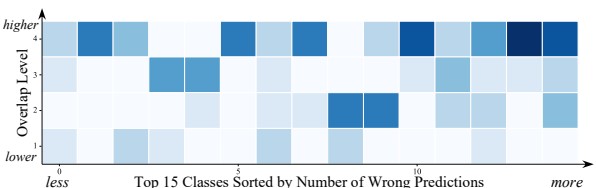

Figure 2: Distribution of wrong predictions across overlap levels. Darker colors indicate more misclassified samples. Under standard training, wrong predictions tend to occur in classes with higher overlap in LIME attribution scores.

2, the majority of misclassifications occur in regions with high overlap levels, i.e., confused classes. This suggests that LIME attribution scores can not solve the error of standard trained model and relying solely on LIME priors may worsen inter-class confusion. This analysis highlights that high-quality attribution priors should not only exhibit strong affinity with the target label but also capture salient and discriminative features critical for distinguishing ambiguous classes.

## 4 THE PROPOSED METHOD

As shown in Figure 3, we introduce an explainable framework that generates comprehensive attribution priors to guide the target model toward a more faithful and robust learning process for answering **Q2**. To capture the salient information for each class, we propose a novel class-aware attribution extraction module to extract priors from large language models automatically. Further we conduct hybrid attribution prior fusion to obtain comprehensive multi-view priors. By leveraging their complementary characteristics, the fused prior highlights both high-affinity frequent tokens and discriminative class-boundary cues. Finally, during explanation-guided learning, we align the target model attribution scores with the fused priors, resulting in a more faithful and robust training.

### 4.1 CLASS-AWARE ATTRIBUTION PRIORS

To address the lack of discriminative attribution signals for ambiguous samples in difficult classes, we propose Class-aware Attribution Priors (CAP) that enhance saliency by emphasizing the key

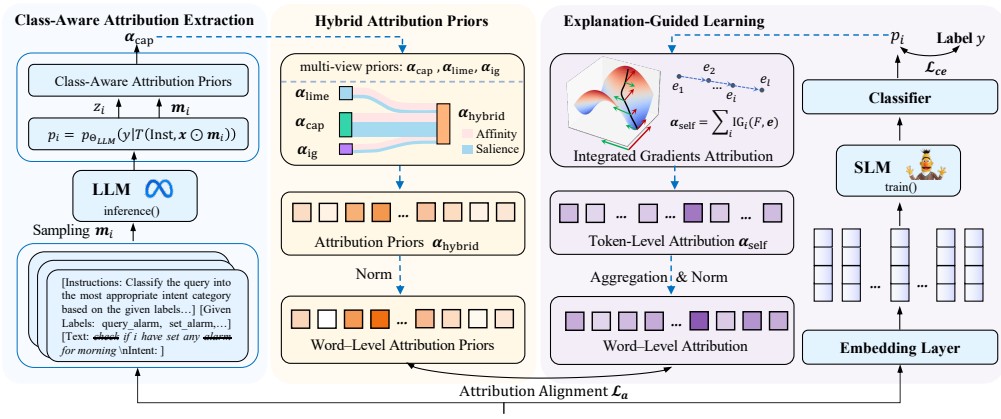

Figure 3: The illustration of our proposed framework. To extract salient priors for ambiguous classes, we first introduce class-aware attribution priors derived from large language models. We then perform hybrid attribution prior fusion to obtain more comprehensive multi-view priors. In the explanation-guided learning module, these priors guide the model toward greater faithfulness and robustness through attribution alignment.

features distinguishing one class from another. As illustrated in Figure 3, we enrich the input text with task instructions and label space, which prompts the LLM to capture the class distribution and produce more salient insights. We then randomly mask individual words and query the LLM for the true-label probability of each masked variant. By using an efficient and stable factorization of the linear equations between these probabilities and the attribution priors, we derive per-word scores.

Given a sentence $\boldsymbol{x} = [w_1, w_2, ..., w_d]$ with $d$ words, we leverage the perturbation vector $\boldsymbol{m} \in \{0, 1\}^d$, where 0 and 1 indicate whether the corresponding word is removed or included, respectively. Due to the significant computational overhead for $2^d$ possible scenarios, we sample $n$ perturbation vectors for predicting the attribution priors $\boldsymbol{\alpha} \in \mathbb{R}^d$. Specifically, to construct $\boldsymbol{m}_i$, we first sample the number of masked words $s$ from the range $[2, \lfloor \frac{d}{2} \rfloor + 1]$. Then we generate the binary mask containing exactly $s$ zeros, where masked positions are chosen uniformly at random. For the sepcific case $s = 1$, each word is masked individually. By repeating this procedure $n$ times, we obtain $\boldsymbol{m}_1, ..., \boldsymbol{m}_n$ to predict and learn the attribution priors $\boldsymbol{\alpha}$.

$$\boldsymbol{\alpha} = \operatorname{argmin}\{\sum_{i=1}^{n}(z_i - \boldsymbol{\alpha}^\mathsf{T}\boldsymbol{m}_i)^2 + \lambda\|\boldsymbol{\alpha}\|^2\}, \tag{1}$$

where $z_i$ is the targeted score, which is positively correlated with the probability of answers, and $\lambda$ is the regularization coefficient. Since probabilities for multi-token answers can be exceedingly small, we apply a logarithmic transformation. Specifically, we set $z_i = -\frac{1}{\log p_i}$, and $p_i$ is computed as follows:

$$p_i = p_{\Theta_{LLM}}(y|T(\texttt{Inst.}, \boldsymbol{x} \odot \boldsymbol{m}_i)), \tag{2}$$

where $T$ is the instruction template, $\texttt{Inst.}$ denotes the task instructions and label space, $\Theta_{LLM}$ is the parameters of the frozen large language model, and $\odot$ denotes the element-wise masking operation.

To solve Eq. (1), we set its derivative to zero and solve for $\boldsymbol{\alpha}$, yielding:

$$\sum_{i=1}^{n}(\boldsymbol{m}_i\boldsymbol{m}_i^\mathsf{T})\boldsymbol{\alpha} + \lambda\boldsymbol{\alpha} = \sum_{i=1}^{n}z_i\boldsymbol{m}_i. \tag{3}$$

The detailed theoretical proof is provided in Appendix A.3. We then construct the matrix form. Let $M = [\boldsymbol{m}_1^\mathsf{T}, \boldsymbol{m}_2^\mathsf{T}, ..., \boldsymbol{m}_n^\mathsf{T}] \in \mathbb{R}^{n \times d}$, $\boldsymbol{z} = [z_1, z_2, ..., z_n] \in \mathbb{R}^d$, the Eq. (3) is:

$$(M^\mathsf{T}M + \lambda I)\boldsymbol{\alpha} = M^\mathsf{T}\boldsymbol{z}. \tag{4}$$

We define $A = M^\mathsf{T} M + \lambda I$, and $\boldsymbol{b} = M^\mathsf{T}\boldsymbol{z}$. We show that $A$ is a symmetric positive definite matrix, and provide the proof in Appendix A.4. Hence, $A$ admits a unique Cholesky factorization (Press, 2007): $A = LL^\mathsf{T}$, where $L$ is a lower triangular matrix with real and positive diagonal entries. To solve $LL^\mathsf{T}\boldsymbol{\alpha} = \boldsymbol{b}$, we then solve the two triangular systems in sequence $L\boldsymbol{y} = \boldsymbol{b}$ and $L^\mathsf{T}\boldsymbol{\alpha} = \boldsymbol{y}$. Since $L$ is a lower triangular matrix, we solve $\boldsymbol{y}$ by forward substitution. Having computed $\boldsymbol{y}$, we solve $\boldsymbol{\alpha}$ by backward substitution because $L^\mathsf{T}$ is a upper triangular matrix. This factorization avoids any explicit matrix inversion and yields $\boldsymbol{\alpha}$ efficiently and stably.

## 4.2 Hybrid Attribution Priors

Each method derives importance scores via distinct mechanisms, yielding its own advantages and shortcomings, as also corroborated by DBLP:conf/nips/HanSL22. To leverage their complementary strengths and enhance overall explanation reliability and model robustness, we incorporate attribution priors from three sources: LIME, our proposed CAP derived from LLMs, and IG from the same model series. Specifically, LIME excels at identifying tokens with high affinity to the target label, CAP emphasizes salient class-boundary terms to distinguish similar classes, and IG captures the internal sensitivity of the model based on gradient information. For fusion, we explore two aggregation strategies: word-wise mean and word-wise max. Before aggregation, all attribution score vectors are normalized to ensure comparability. Formally, given attribution priors $\boldsymbol{\alpha}_{\text{lime}}$, our proposed CAP priors $\boldsymbol{\alpha}_{\text{cap}}$, and IG signals $\boldsymbol{\alpha}_{\text{ig}}$, we define the fused prior as

$$\boldsymbol{\alpha}_{\text{hybrid}} = \texttt{aggregate}\left(\{(\boldsymbol{\alpha}_{\text{lime}}), (\boldsymbol{\alpha}_{\text{cap}}), (\boldsymbol{\alpha}_{\text{ig}})\}\right) \tag{5}$$

where `aggregate` denotes either the element-wise mean or max operation. This results in a comprehensive word-level attribution prior that maintains full coverage while integrating multi-source rationale signals.

## 4.3 Explanation-Guided Learning

After obtaining attribution priors, we conduct word-level attribution alignment between the priors and the target model self-attribution. Specifically, we leverage integrated gradients to attribute the prediction of the target model to input features. Formally, given an input sentence $\boldsymbol{x} = [\boldsymbol{e}_1, \boldsymbol{e}_2, ..., \boldsymbol{e}_{d'}]$ consisting of $d'$ tokens, $\boldsymbol{e}_i \in \mathbb{R}^l$ is the $i$-th token embedding encoded by the embedding layer, and $l$ is the dim of token embedding. For each feature $\boldsymbol{e}$, $\boldsymbol{e}'$ is a baseline feature, and the target model is $F(\cdot)$. IG estimates the contribution of each input by integrating the gradients of $F$ along the straight-line path from $\boldsymbol{e}'$ to $\boldsymbol{e}$ as the integral path:

$$F(\boldsymbol{e}) - F(\boldsymbol{e}') = \sum_{i=1}^{l} \text{IG}_i(F, \boldsymbol{e}) = \sum_{i=1}^{l} [(e_i - e_i') \times \int_{\mu=0}^{1} \frac{\partial F(\boldsymbol{e}' + \mu \times (\boldsymbol{e} - \boldsymbol{e}'))}{\partial e_i} \mathrm{d}\mu]. \tag{6}$$

Then IG attribution scores $\boldsymbol{\alpha}_{\text{self}}$ is normalized for the subsequent alignment process. For clarity, let $\boldsymbol{a}^p$ denote the normalized external attribution priors, and $\boldsymbol{a}^s$ represent the normalized self-attribution scores from the target model. We use Mean Squared Error (MSE) as the distance metric for alignment:

$$\mathcal{L}_a = \|\boldsymbol{a}^p - \boldsymbol{a}^s\|^2. \tag{7}$$

For overall objective, we jointly optimize the standard vanilla cross-entropy loss between the model predictions and the ground-truth labels, and our attribution-driven objective to train the compact model. The overall objective is defined as:

$$\mathcal{L} = \mathcal{L}_{ce} + \beta\mathcal{L}_a, \tag{8}$$

where $\mathcal{L}_{ce} = -\frac{1}{N}\sum_{i=1}^{N} y_i \log p_{\theta_s}(y_i|\boldsymbol{x}_i)$ is the cross-entropy loss function, $N$ is the number of training samples and $\beta$ is the hyperparameter that balances attribution guidance against the standard classification objective.

## 5 Experiments

### 5.1 Datasets and Baselines

We evaluate on three intent classification datasets: HWU64 (Liu et al., 2019a), Banking77 (Casanueva et al., 2020) and Clinc150 (Larson et al., 2019). For the baseline, we select two methods

| Methods | HWU64 | | | | Banking77 | | | | Clinc150 | | | |
|---|---|---|---|---|---|---|---|---|---|---|---|---|
| | Acc | Acc$_{AD}$ | Com↑ | Suf↓ | Acc | Acc$_{AD}$ | Com↑ | Suf↓ | Acc | Acc$_{AD}$ | Com↑ | Suf↓ |
| *Baseline* | | | | | | | | | | | | |
| Roberta-Base | 75.56 | 62.36 | 65.52 | 34.97 | 74.04 | 53.82 | 67.06 | 38.30 | 87.99 | 71.70 | 65.38 | 30.91 |
| *Hybrid size = 1* | | | | | | | | | | | | |
| IG | 77.04 | 65.80 | 68.93 | 32.67 | **79.09** | 56.57 | 68.05 | 36.79 | 89.58 | 72.29 | 67.09 | 26.36 |
| LIME | 77.13 | 66.96 | 68.79 | 31.53 | 78.50 | 56.96 | 73.21 | 33.19 | 89.24 | 74.49 | 71.23 | 26.68 |
| CAP | 76.57 | 66.31 | 68.35 | 29.76 | 78.83 | 57.24 | 72.95 | **30.98** | 89.20 | 73.39 | 65.69 | 26.35 |
| *Hybrid size = 2* | | | | | | | | | | | | |
| IG+LIME | 77.88 | 66.40 | 66.86 | 25.49 | 78.57 | 57.12 | 72.33 | 31.83 | 89.64 | 73.38 | 67.50 | 26.97 |
| CAP$_{+IG}$ | 78.16 | 66.91 | 66.56 | 31.16 | 78.89 | 57.17 | 69.42 | 37.02 | 89.58 | 74.16 | 69.95 | 27.76 |
| CAP$_{+LIME}$ | 77.50 | 67.38 | 69.12 | **24.55** | 78.31 | 57.32 | 73.87 | 31.70 | 89.00 | 74.19 | **73.66** | **24.34** |
| *Hybrid size = 3* | | | | | | | | | | | | |
| CAP$_{Hybrid}$ | **79.09** | **70.63** | **69.17** | 30.66 | 78.63 | **58.05** | **73.97** | 31.57 | **89.69** | **74.70** | 70.59 | 25.53 |

Table 1: Results on three datasets in the 5-shot setting. The *Hybrid Size* here represents the supervision attribution composed of several attributions combined together, e.g. CAP$_{+LIME}$ means that we merged attribution from CAP and LIME. The **best** and second-best results in each column are shown in **bold** and underlined.

with different principles commonly used in explanation guided learning for comparison: Integrated Gradients (IG) (Sundararajan et al., 2017) and LIME (Ribeiro et al., 2016). Further details are presented in Appendix A.2.

## 5.2 ADVERSARIAL DATASET CONSTRUCTION

To assess the robustness of classifiers trained on the specific datasets, we automatically generate adversarial test sets using two complementary attack strategies: keyword addition and keyword replacement. *Keyword Addition:* Inject one or more keywords from an adversarial class into each original test example. The true label remains unchanged, so a robust model should still predict the original class. *Keyword Replacement:* Replace keywords from the original class with those from the adversarial class, aiming to preserve fluency while misleading the model toward the adversarial label. In the adversarial class selection, we first applied our method CAP to calculate the word-level attribution scores of training dataset for each class. These scores are subsequently aggregated to obtain a class-level keyword list. Pairwise difficulty is quantified by measuring the overlap of keyword distributions between classes. For every source class, we randomly sample $n$ adversarial target classes; the value of $n$ is determined by the total number of classes in the dataset (1 for HWU64, 2 for Banking77, and 3 for Clinc150). We construct this adversarial dataset by using GPT-4o, the generation prompt templates are provided in Appendix A.7.

## 5.3 IMPLEMENTATION DETAILS

For evaluation metrics, we report accuracy to assess overall performance for full-data, few-shot and adversarial settings. Additionally, following (DeYoung et al., 2020), we report comprehensiveness and sufficiency to quantify how well the model explanations reflect the decision process. Specifically, given a sample $x_i$, the comprehensiveness measures the drop in confidence when the rationale $r_i$ is removed.

$$\text{Com} = \frac{p_{\theta_s}(y_i|x_i) - p_{\theta_s}(y_i|x_i \setminus r_i)}{p_{\theta_s}(y_i|x_i)}, \tag{9}$$

where $y_i$ denotes the predicted label of $x_i$, and $p_{\theta_s}(y_i|x_i)$ is the probability that model $\theta_s$ assigns to $y_i$ given input $x_i$. A higher comprehensiveness score indicates that $r_i$ is critical to the model decision, whereas a lower score implies that the rationale has little influence. Conversely, sufficiency quantifies how well the extracted rationale alone supports the prediction.

$$\text{Suf} = \frac{p_{\theta_s}(y_i|x_i) - p_{\theta_s}(y_i|r_i)}{p_{\theta_s}(y_i|x_i)}. \tag{10}$$

| Method | Accuracy ↑ | | Com ↑ | Suf ↓ |
|---|---|---|---|---|
| | Test | Adversarial | | |
| Base | 92.63 | 70.29 | 56.73 | 35.87 |
| IG | 93.02 | 77.91 | 61.75 | 32.81 |
| LIME | 92.82 | 75.85 | 62.87 | 32.23 |
| CAP | 93.41 | 78.20 | 68.00 | 24.30 |
| $CAP_{Hybrid}$ | **93.51** | **84.97** | **68.21** | **23.98** |

Table 2: Performance on Banking77 under the full-train setting. The **best** and second-best results in each column are shown in **bold** and underlined. *Com* and *Suf* stand for the comprehensiveness and sufficiency calculated on test set respectively.

A smaller sufficiency score, that is a minimal drop in confidence when using only $r_i$, indicates that the rationale by itself is sufficient for predicting $y_i$.

For parameter settings, we use RoBERTa-Base as the standard small language model for both training and extracting Integrated Gradients (IG) attributions. For extracting attribution scores using LIME and our proposed CAP method, we consistently adopt Llama-3-8B-Instruct as the large language model. Hyperparameters for training and the selection of aggregation can be found in the Appendix A.5 and A.6.

## 5.4 RESULT ANALYSIS

To answer **Q3**, we conduct comprehensive experiments and report the evaluation results in both *few-shot* and *full-data* training scenarios.

### 5.4.1 FEW-SHOT RESULTS

To evaluate whether our hybrid attribution prior can facilitate more reliable reasoning under low-resource conditions, we conduct experiments on 5-shot subsets of all three datasets. Table 1 reports the detailed results that on all of three datasets, single-prior (IG, LIME, and CAP) outperform the RoBERTa-Base baseline across all evaluation metrics. Each method exhibits unique strengths: IG consistently improves classification accuracy (e.g., 79.09% on Banking77), suggesting it helps the model focus on decision-critical tokens. LIME, by contrast, yields the highest comprehensiveness in most cases (e.g., 71.23% on Clinc150), indicating that its token-level perturbation captures prior more effectively. CAP achieves the best sufficiency score (29.76% on HWU64), showing that it guides the model to make predictions based on compact and essential evidence. This also illustrates our point of view that these attribution methods show different preferences in not only attribution mechanisms but also in the supervision of downstream tasks.

In particular, combining these priors leads to a synergistic effect. Hybrid variants generally outperform their individual counterparts, suggesting the complementarity of different attribution strategies. Among them, $CAP_{+LIME}$ achieves the lowest sufficiency and high comprehensiveness, which validates the effectiveness of our proposed CAP. Our final method, $CAP_{Hybrid}$, which integrates CAP, IG, and LIME priors, achieves the highest overall accuracy (79.09%) in HWU64 dataset. Simultaneously, it maintains strong interpretability, with a comprehensiveness score of 69.12% and a sufficiency score of 30.66%, both ranking second among all methods. This suggests that $CAP_{Hybrid}$ not only improves prediction accuracy but also promotes more faithful and stable reasoning processes. In the in-domain adversarial setting, the RoBERTa-Base baseline exhibits a substantial performance drop compared to the original test set, highlighting the vulnerability of models under standard training. In contrast, $CAP_{Hybrid}$ significantly improves adversarial accuracy across all three datasets, with notable gains on HWU64 (8%) and Banking77 (4%). These results demonstrate that our method enhances robustness of models in more challenging settings.

### 5.4.2 FULL TRAINING RESULTS

Table 2 reports the results on Banking77 under the full-data training setting. As shown, our method continues to demonstrate strong performance. $CAP_{Hybrid}$ achieves the highest test accuracy

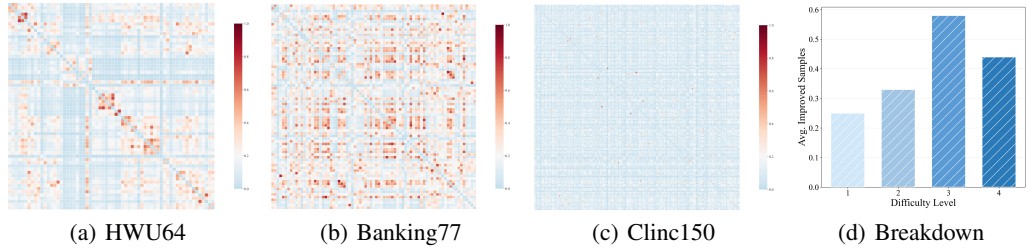

(a) HWU64      (b) Banking77      (c) Clinc150      (d) Breakdown

Figure 4: Subfigure (a)-(c) show the class similarity of HWU64, Banking77 and Clinc150 datasets. Darker colors denote higher similarities. Subfigure (d) shows the imporvements performance breakdown across difficulty levels in 5-shot Banking77 dataset.

93.51%(vs 92.63%) and adversarial accuracy 84.97%(vs 70.29%), along with the best comprehensiveness 68.21%(vs 56.73%) and sufficiency 23.98%(vs 23.98%). These results confirm that our attribution-guided supervision remains effective even when ample training data is available.

### 5.5 QUALITATIVE ANALYSIS OF DATASET PROPERTIES AND METHOD EFFECTIVENESS

To further investigate the factors affecting the effectiveness of our method and to better understand the conditions under which attribution priors offer the greatest benefit, we conduct qualitative analyses centered on two key observations. First, Figure 4(d) illustrates the relationship between the number of corrected cases achieved by our attribution-guided training and the lexical overlap between the ground-truth labels and the model's original incorrect predictions—a proxy for instance difficulty. The results reveal that our method is particularly effective in handling challenging examples (difficulty levels 3 and 4), where the confusing classes share a high degree of token-level similarity. This highlights the model's enhanced ability to tease apart subtle semantic distinctions that are often lost under standard training.

Second, we analyze the overall structure of each dataset by computing the mean sentence embedding for each class and measuring pairwise class similarities. As shown in the similarity heatmaps Figure 4(a)-(c), Banking77 and HWU64 contain clusters of highly similar classes, indicating the presence of *inherent difficulty imbalance*. In contrast, Clinc150 exhibits more uniformly distributed and well-separated class representations, with less inter-class confusion. This distinction clarifies our empirical findings: on HWU64 and Banking77, baseline models frequently misclassify these fine-grained categories due to spurious token co-occurrence patterns. Our approach, by aligning predictions with richer and more discriminative attribution priors, mitigates this issue and enables the model to focus on features that are more indicative of class-level meaning rather than surface lexical overlap. In contrast, for Clinc150, even simple priors derived from LIME suffice to provide meaningful guidance, thereby limiting the room for further gains. Consequently, our method achieves marginal improvements in that setting. These findings suggest that our attribution-based supervision not only improves robustness to lexical ambiguity but also enhances the model's capacity for deeper semantic reasoning, particularly in challenging scenarios involving closely related or fine-grained classes.

## 6 CONCLUSION

We identify a critical limitation in existing attribution guided learning: when categories exhibit high semantic overlap, models trained with surface-level priors often fail to distinguish fine-grained classes. To address this, we propose Class-Aware Attribution Priors (CAP), which inject task-specific and label-informed semantic constraints into model training, encouraging more meaningful feature attribution. We further introduce $\text{CAP}_{Hybrid}$, a fusion approach that integrates multiple complementary attribution sources to form richer, more balanced priors. Our method consistently enhances performance, interpretability, and robustness across various datasets, offering a promising direction toward building more transparent and reliable models.

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

# A APPENDIX

## A.1 THE USE OF LARGE LANGUAGE MODELS

During the preparation of this paper, we used large language models (LLMs) solely as general-purpose writing assistants. Specifically, LLMs were employed to help refine the clarity, grammar, and readability of our drafts, as well as to suggest alternative phrasings in English. Importantly, all conceptual contributions including the design of research questions, development of methods, execution of experiments, and interpretation of results were conceived and carried out entirely by the authors. The authors carefully reviewed and edited all text suggested by LLMs to ensure accuracy and originality, and we take full responsibility for the final content of the paper.

## A.2 DATASETS AND BASELINES

We evaluate on three multi-domain intent classification datasets, each covering diverse real-world scenarios: (1) HWU64 (Liu et al., 2019a) contains 64 intents drawn from 18 everyday domains (e.g., alarm, cooking, transportation), with utterances balanced across intents. (2) Banking77 (Casanueva et al., 2020) comprises 13,083 labeled customer utterances spanning 77 fine-grained intents in the banking domain. (3) Clinc150 (Larson et al., 2019) offers 22,500 user utterances evenly distributed over 150 intents in 10 general domains, facilitating robust cross-domain evaluation.

For the baseline, we select two methods with different principles commonly used in explanation guided learning for comparison: Integrated Gradients (IG) (Sundararajan et al., 2017) We first train a model using only the classification loss and then compute IG attribution scores on the training set using this model. These attribution scores are subsequently incorporated as supervision signals to train a new model from scratch. LIME (Ribeiro et al., 2016) We employ LIME to compute attribution scores on the training dataset from large language model, and use these scores as external supervision to guide the training of the smaller model.

## A.3 PROOF OF EQ. (3)

This section describes the proof of Eq. (3). For clarity, we first restate our objective, i.e. Eq. (1):

$$\boldsymbol{\alpha} = \operatorname{argmin}\{\sum_{i=1}^{n}(z_i - \boldsymbol{\alpha}^\mathsf{T}\boldsymbol{m}_i)^2 + \lambda\|\boldsymbol{\alpha}\|^2\}, \tag{11}$$

where $\boldsymbol{\alpha} \in \mathbb{R}^d$ is the attribution parameter vector to be estimated, each $\boldsymbol{m}_i \in \mathbb{R}^d$ is a mask vector with associated score $z_i$, and $\lambda > 0$ is the regularization coefficient. To find the minimizer, we calculate its derivative. Specifically, let $\mathcal{L}(\boldsymbol{\alpha})$ be the function,

$$\mathcal{L}(\boldsymbol{\alpha}) = \sum_{i=1}^{n}(z_i - \boldsymbol{\alpha}^\mathsf{T}\boldsymbol{m}_i)^2 + \lambda\boldsymbol{\alpha}^\mathsf{T}\boldsymbol{\alpha}. \tag{12}$$

The derivative of $\mathcal{L}(\boldsymbol{\alpha})$ with respect to $\boldsymbol{\alpha}$ is as follows,

$$\nabla_{\boldsymbol{\alpha}}\mathcal{L} = -2\sum_{i=1}^{n}(z_i - \boldsymbol{\alpha}^\mathsf{T}\boldsymbol{m}_i)\boldsymbol{m}_i + 2\lambda\boldsymbol{\alpha}. \tag{13}$$

To find the minimizer, we set $\nabla_{\boldsymbol{\alpha}}\mathcal{L} = 0$, yielding

$$\sum_{i=1}^{n}(\boldsymbol{\alpha}^\mathsf{T}\boldsymbol{m}_i)\boldsymbol{m}_i + \lambda\boldsymbol{\alpha} = \sum_{i=1}^{n}z_i\boldsymbol{m}_i. \tag{14}$$

Next we transform the left-hand side of Eq. (14). For $(\boldsymbol{\alpha}^\mathsf{T}\boldsymbol{m}_i)\boldsymbol{m}_i$ item, let $t = \boldsymbol{\alpha}^\mathsf{T}\boldsymbol{m}_i$, which is a scalar. Then $t\boldsymbol{m}_i = \boldsymbol{m}_i t$ and $\boldsymbol{\alpha}^\mathsf{T}\boldsymbol{m}_i = \boldsymbol{m}_i^\mathsf{T}\boldsymbol{\alpha}$. We have

$$\begin{aligned}(\boldsymbol{\alpha}^\mathsf{T}\boldsymbol{m}_i)\boldsymbol{m}_i &= \boldsymbol{m}_i t \\ &= \boldsymbol{m}_i(\boldsymbol{m}_i^\mathsf{T}\boldsymbol{\alpha}) \\ &= (\boldsymbol{m}_i\boldsymbol{m}_i^\mathsf{T})\boldsymbol{\alpha}.\end{aligned}$$

Substituting $(\boldsymbol{\alpha}^\mathsf{T}\boldsymbol{m}_i)\boldsymbol{m}_i = (\boldsymbol{m}_i\boldsymbol{m}_i{}^\mathsf{T})\boldsymbol{\alpha}$ into Eq. (14) gives

$$\sum_{i=1}^{n}(\boldsymbol{m}_i\boldsymbol{m}_i{}^\mathsf{T})\boldsymbol{\alpha} + \lambda\boldsymbol{\alpha} = \sum_{i=1}^{n} z_i\boldsymbol{m}_i, \tag{15}$$

which is exactly Eq. (3).

## A.4 PROOF OF POSITIVE DEFINITE MATRIX A

We now show that
$$A = M^\mathsf{T}M + \lambda I, \tag{16}$$
is both symmetric and positive definite, where $I$ is the identity matrix and $\lambda > 0$.

Symmetry.

$$\begin{aligned}
A^\mathsf{T} &= (M^\mathsf{T}M + \lambda I)^\mathsf{T} \\
&= (M^\mathsf{T}M)^\mathsf{T} + (\lambda I)^\mathsf{T} \\
&= M^\mathsf{T}M + \lambda I^\mathsf{T} \\
&= A.
\end{aligned}$$

Positive Definiteness. For any nonzero real column vector $\boldsymbol{x}$, we have

$$\begin{aligned}
\boldsymbol{x}^\mathsf{T}A\boldsymbol{x} &= \boldsymbol{x}^\mathsf{T}(M^\mathsf{T}M + \lambda I)\boldsymbol{x} \\
&= \boldsymbol{x}^\mathsf{T}(M^\mathsf{T}M)\boldsymbol{x} + \boldsymbol{x}^\mathsf{T}(\lambda I)\boldsymbol{x} \\
&= (\boldsymbol{x}^\mathsf{T}M^\mathsf{T})(M\boldsymbol{x}) + \lambda\boldsymbol{x}^\mathsf{T}\boldsymbol{x} \\
&= \|M\boldsymbol{x}\|^2 + \lambda\|\boldsymbol{x}\|^2.
\end{aligned}$$

Since $\|M\boldsymbol{x}\|^2 \geq 0$ for all $\boldsymbol{x}$ and $\lambda\|\boldsymbol{x}\|^2 > 0$ whenever $\boldsymbol{x} \neq \boldsymbol{0}$, it follows that

$$\boldsymbol{x}^\mathsf{T}A\boldsymbol{x} > 0, \quad \forall \boldsymbol{x} \neq \boldsymbol{0}, \tag{17}$$

thus $A$ is positive definite.

## A.5 TRAINING PARAMETER SETTING

When calculating our class-aware arribution priors, we set the $\lambda = 0.1$ and $n = 100$ in Eq. (1). And when training the Roberta-Base model, we set the learning rate to 5e-5, batch size to 16, and use early stopping with a patience of 10. Additionally, we apply gradient norm clipping with a maximum norm of 1.0. For our attribution-aligned training, we experiment with attribution loss weighting coefficient $\beta = 1$. While both *mean* and *max* aggregation strategies were considered for hybrid prior methods, we report only the test results corresponding to the combination that achieved the best validation performance. The comparison experiment can be found in Appendix A.6.

## A.6 THE COMPARSION OF DIFFERENT AGGREGATION OPERATION

We compare different aggregation strategies in terms of their impact on classification accuracy. As shown in Table 3, the mean aggregation achieves the best predictive performance in most cases.

## A.7 ADVERSARIAL DATASET GENERATION PROMPT

In this section, we present all the prompts utilized for constructing the adversarial datasets.

| Dataset | Banking77 | | Clinc150 | | HWU64 | |
|---|---|---|---|---|---|---|
| Aggretion | Mean | Max | Mean | Max | Mean | Max |
| IG+LIME | **78.57** | 76.91 | **89.64** | 89.44 | **77.88** | 77.53 |
| CAP+IG | 78.81 | **78.89** | 89.58 | 89.31 | **78.16** | 78.04 |
| CAP+LIME | 77.89 | **78.31** | 89.00 | 88.71 | **77.50** | 77.29 |
| $CAP_{Hybrid}$ | **78.63** | 78.47 | **89.69** | 88.98 | 77.41 | **79.09** |

Table 3: The comparision of accuracy for different aggregation methods on the 5-shot setting.

```
Task:  Generate a keyword addition adversarial example for the following
text by adding keywords from a adversarial class without changing the
classification result.
Original text:  {original text}
Original class:  {original label}
Adversarial class:  {adversarial label}
Adversarial class keywords:  {adversarial keywords}
Adversarial class example sentences:  {adversarial sentence}
Requirements:
1.  Preserve the original meaning as much as possible
2.  Add 1{2 keywords from the source class, but avoid making the intent
overly explicit
3.  Ensure the new text is still classified as the target class
4.  Ensure the text remains natural and fluent
5.  Only return the modified text, nothing else
Adversarial example:
```

Table 4: The template of the prompt we used for generate *Keyword Addition* adversarial examples.

```
Task:  Generate a keyword replacement adversarial example by replacing
original class keywords in the following text with keywords from a
adversarial class.
Original text:  {original text}
Original class:  {original label}
Adversarial class:  {adversarial label}
Adversarial class keywords:  {adversarial keywords}
Adversarial class example sentences:  {adversarial sentence}
Requirements:
1.  Replace target keywords with suitable original-class keywords, and
optionally add new ones
2.  Preserve the original sentence structure (e.g., questions,
conjunctions)
3.  Ensure the new text remains semantically coherent and realistic
4.  Ensure the modified text is classified into the new target class
5.  Only return the modified text, nothing else
Adversarial example:
```

Table 5: The template of the prompt we used for generate *Keyword Replacement* adversarial examples.

