# OpenReview forum: "Hybrid Attribution Priors for Explainable and Robust Model Training"
_ICLR.cc/2026/Conference — ICLR 2026 Conference Withdrawn Submission_

### Official Review · Reviewer_v41g · 2025-10-28

**Soundness:** 3
**Presentation:** 2
**Contribution:** 2
**Rating:** 2
**Confidence:** 4

**Summary:**

The paper proposes a large language model (LLM)-based feature attribution mechanism called Class-Aware Attribution Prior (CAP). CAP can be integrated with existing attribution methods such as Integrated Gradients (IG) and LIME to enhance model interpretability and robustness.

**Strengths:**

1. The proposed LLM-based feature attribution method is novel and offers a promising direction for future research and exploration.
2. The hybrid approach is conceptually sound and empirically validated, demonstrating that LLM-based feature attribution provides complementary benefits to traditional SLM-based methods.

**Weaknesses:**

1. The aggregation mechanism could be further investigated. In the simulation study, the authors did not specify which aggregation method (mean or max) was used. Including an ablation study comparing different aggregation strategies would make the analysis more comprehensive.
2. The formulation of the LLM attribution priors (both in Equation (1) and the subsequent algorithm, ranging lines 254–276) is equivalent to Ridge regression. The authors may consider citing relevant references and leveraging existing solvers for completeness and clarity.

**Questions:**

The writing quality could be improved, particularly in the citation formatting. Several citations appear broken (e.g., displaying “DBLP” or showing \citet instead of \citep around line 155). Also, subsubsections commands can be replaced to avoid X.0.Y section title in the third section.

---

### Official Review · Reviewer_nbyJ · 2025-10-28

**Soundness:** 1
**Presentation:** 2
**Contribution:** 2
**Rating:** 2
**Confidence:** 5

**Summary:**

The work proposes Hybrid Attribution Priors (CAPHybrid), a framework designed to improve small language models through attribution-guided learning. The work mentions that attribution methods such as Integrated Gradients and LIME often produce non-discriminative explanations, emphasizing overlapping tokens across semantically similar classes. To address this, they introduce Class-Aware Attribution Priors (CAP), which use large language models to generate class-sensitive attribution signals, and further combine CAP with traditional attribution methods to form a hybrid prior that provides more balanced supervision. CAPHybrid guides model training by aligning the model’s internal attributions with these priors. Experiments on three intent-classification datasets (HWU64, Banking77, and Clinc150) under full-data, few-shot, and adversarial settings show modest but consistent improvements in accuracy, attribution consistency, and robustness to adversarial perturbations.

**Strengths:**

- The paper is generally well-written and structured, guiding the reader from problem motivation to method formulation and evaluation.
- The paper provides both quantitative and qualitative analyses, as well as an adversarial evaluation protocol to assess robustness.
- The paper evaluates CAPHybrid under diverse conditions including full-data, few-shot, and adversarial settings.

**Weaknesses:**

- The paper argues that existing attribution methods such as LIME and IG fail to produce class-aware or discriminative explanations, but it does not compare against more advanced interpretability methods that have explicitly addressed this limitation. For instance, transformer-based attribution approaches [1,2,3] provide **stronger baselines** for evaluating “class-aware” interpretability. Without these comparisons, it remains unclear whether CAPHybrid truly advances beyond the state of the art in explanation quality.

- The method’s effectiveness is primarily validated **only** through the downstream performance of SLM trained with CAPHybrid guidance. However, the paper does not directly assess whether the proposed attribution priors are trustworthy or truthful. There is no evaluation of the priors themselves against human rationales from external benchmarks such as ERASER [4] which limits confidence that the priors reflect genuine causal reasoning rather than spurious patterns.

- The improvements reported across accuracy, robustness, and explanation consistency are generally **modest**. While consistent, these gains are small relative to the added complexity of generating and combining multiple attribution sources. This suggests that CAPHybrid provides incremental rather than substantial benefits, and its practical significance may be limited.

- The paper equates improvements in Comprehensiveness and Sufficiency with enhanced interpretability, but these metrics only measure internal consistency (faithfulness) rather than genuine human interpretability. Without human-grounded or causal validation, it is unclear whether the model’s explanations are actually more understandable or truthful. Thus, **the claim of improved interpretability** may be overstated relative to what the reported evidence supports.

---
[1] Hila Chefer, Shir Gur, and Lior Wolf. Transformer interpretability beyond attention visualization. CVPR 2021.

[2] Hila Chefer, Shir Gur, and Lior Wolf. Generic attention-model explainability for interpreting bi-modal and encoder-decoder transformers. ICCV 2021.

[3] J. Chen, X. Li, L. Yu, D. Dou, and H. Xiong. Beyond intuition: Rethinking token attributions inside transformers. Transactions on Machine Learning Research (TMLR), 2022.

[4] Jay DeYoung, Sarthak Jain, Nazneen Fatema Rajani, Eric Lehman, Caiming Xiong, Richard Socher, and Byron C. Wallace. ERASER: A benchmark to evaluate rationalized NLP models. ACL 2020

**Questions:**

- Could the authors clarify why they did not include comparisons against more advanced attribution or class-aware interpretability methods? It would be helpful to understand whether CAPHybrid offers complementary benefits or improvements in these contexts.

- Generating LLM-based CAP priors and combining them with IG/LIME appears computationally expensive. Could the authors provide details on the computational cost, inference latency, or resource overhead? How scalable is CAPHybrid to larger datasets or domains beyond intent classification?

- The experiments are limited to three intent classification datasets with relatively short texts. Have the authors tested or considered applying CAPHybrid to longer-text or multi-hop reasoning datasets to assess whether the framework generalizes to more complex interpretability settings?

- The paper leverages rationales or attribution signals generated by a large language model (LLM) as supervision for training small models. However, LLM-generated explanations are not guaranteed to be faithful or causally aligned with actual decision processes. If the LLM explanations themselves are imperfect, might the SLMs risk inheriting or amplifying those biases?

---

### Official Review · Reviewer_15EW · 2025-10-29

**Soundness:** 2
**Presentation:** 3
**Contribution:** 2
**Rating:** 2
**Confidence:** 4

**Summary:**

This paper proposes a pipeline to supervise small language models not just on what label to predict, but on why, using attribution priors derived from a larger model. The core idea is: generate per-word importance scores (CAP) by masking tokens and probing a frozen LLM, optionally fuse them with other attribution methods (CAPHybrid), and then train the student model so that its own integrated-gradients map matches that prior, alongside normal cross-entropy.

**Strengths:**

The paper targets a practical goal: making small language models both accurate and interpretable.

The work provides a clear way to get discriminative supervision without manual rationales.

**Weaknesses:**

The fundamental issue is that the paper’s core prior (CAP) is built by repeatedly querying a large language model with masked versions of each training example, then fitting a regression to infer per-word importance. This seems extremely expensive at realistic dataset sizes, especially for long texts. On top of that, CAPHybrid, which is the best performing method, fuses multiple attribution sources (CAP, LIME, IG), some of which are themselves multi-pass methods.

There are claims of broad robustness, but it is measured using synthetic keyword-style attacks designed around the paper’s target failure mode. No real distribution shift or harder tasks are tested.

The paper relies on comprehensiveness and sufficiency to claim better interpretability, but these metrics are easy to game and don’t prove faithfulness. Amongst numerous issues with these metrics, the model during training is used to explicitly align its attributions to a fixed prior, so high comprehensiveness and low sufficiency mainly show that the model was forced to depend on those highlighted tokens.

The  LLM is asked how confident it is in a given label, even after words are removed, it may be pressured to justify that label using whatever weak hints are left. That can mark tokens as “important” even if they don’t truly distinguish that label from others, so the student may learn to copy superficial justifications instead of real decision cues.

**Questions:**

N/A

---

### Official Review · Reviewer_PjFK · 2025-11-03

**Soundness:** 2
**Presentation:** 2
**Contribution:** 2
**Rating:** 4
**Confidence:** 4

**Summary:**

This paper addresses the challenge of training interpretable and robust small language models (SLMs) for text classification through explanation-guided learning. The authors first identify fundamental limitations in some attribution methods (LIME, SHAP, Integrated Gradients): namely, "homogenization" (producing highly correlated scores) and "class confusion" (focusing on shared keywords rather than discriminative features between similar classes). To address these issues, the paper proposes Class-Aware Attribution Prior (CAP), which extracts discriminative attribution signals from large language models via an ablation procedure, enriching inputs with task instructions and label space information. The authors then introduce CAPHybrid, which fuses attribution priors from CAP, LIME, and IG to capture complementary perspectives. During training, SLMs are guided to align their self-attributions (computed via Integrated Gradients) with these hybrid priors through an attribution alignment loss combined with standard cross-entropy loss. Experiments across three intent classification datasets (HWU64, Banking77, Clinc150) show that CAPHybrid consistently improves classification accuracy, interpretability metrics (comprehensiveness and sufficiency), and adversarial robustness. The method proves particularly effective on datasets with high inter-class similarity, achieving notable adversarial accuracy gains (e.g., 14.7% improvement on Banking77 in the full-data setting).

**Strengths:**

* Originality: The paper highlights homogenization and class confusion in methods such as LIME and SHAP when applied to language model predictions. The efficient Cholesky factorization solution to the attribution optimization problem (Equations 1-4) demonstrates technical innovation and can potentially be adopted to speed up future perturbation-based attribution techniques.
* Quality: The evaluation focuses on the real-world task of intent classification, employing three diverse datasets, although lacking breadth in the model dimension. The adversarial robustness evaluation is particularly rigorous, and systematic ablations (Table 1) demonstrate the value of each attribution component.
* Clarity: Figure 3 provides an effective overview of the proposed approach, and the formulation in Section 4.1 is sound. The distinction between affinity-based (LIME) and salience-based (CAP) attribution is clearly articulated, though the exact mechanism by which CAP resolves class confusion could benefit from additional intuitive explanation.
* Significance: The insight that attribution-guided training is most beneficial for datasets with high inter-class similarity (Banking77, HWU64) versus well-separated classes (Clinc150) provides valuable guidance for practitioners, motivating the potential inclusion of attribution priors in common training practices.

**Weaknesses:**

In the related work section, authors correctly identify "[...] the elicitation of more reliable explanations as a key pathway to enhance interpretative robustness". However, the investigation focus solely on methods such as SHAP, LIME and IG that were developed long before present-day language models, and importantly not specifically attuned to the language domain. Much research was conducted on the development of new methods that would reflect input contributions more faithfully in these domain, for example, by decomposing model components' contributions across layers [1], designing custom gradient propagation rules to extend the default LRP technique to the Transformer architecture [2] or exploiting attention-weighted value vectors to quantify context mixing between model layers [3]. The current answer to the first research question currently relies on a single established white-box method (IG), but ignores the aforementioned improvements (and many other, see e.g. Section 3.1 in [4] for a review), or the commonplace use of contrastive attribution targets [5] when attributing language models' predictions. In light of this, the answer to Q1 cannot be considered appropriate in its current state, since the current selection of method is not representative of reliable attribution approaches for LMs. Moreover, the use of contrastive explanations formulated by [5] can already mitigate the problem of class confusion, and could have been used as a baseline to the proposed CAP method. While white-box methods might be impractical to run on larger models, the focus on SLMs of this paper should greatly simplify such an evaluation.

From a methodological perspective, the class-aware perturbation proposed by the authors does not account for the implicit structure present in language. In the case of masking one word at a time, for example, multi-word expressions would never be entirely masked for the model, and in some cases masking function words that carry low semantic meaning could still change the meaning of a sentence entirely. Recent work considers this constraint when designing perturbations [6], resulting in more faithful and coherent explanations. In terms of evaluation, using soft variants of Comprehensiveness and Sufficiency should be preferable to estimate attribution faithfulness [7]. Moreover, the use of a single model combination (RoBERTa + LLaMA) in the experimental evaluation does not provide sufficient evidence for the generality of the findings.

**References:**

- [1]: [DecompX: Explaining Transformers Decisions by Propagating Token Decomposition](https://aclanthology.org/2023.acl-long.149/) (Modarressi et al., ACL 2023)
- [2] [AttnLRP: attention-aware layer-wise relevance propagation for transformers](https://dl.acm.org/doi/10.5555/3692070.3692076) (Achtibat et al., ICML 2024)
- [3]: [Explaining How Transformers Use Context to Build Predictions](https://aclanthology.org/2023.acl-long.301/) (Ferrando et al., ACL 2023)
- [4]: [A Primer on the Inner Workings of Transformer-based Language Models](https://arxiv.org/abs/2405.00208) (Ferrando et al. 2024)
- [5]: [Interpreting Language Models with Contrastive Explanations](https://aclanthology.org/2022.emnlp-main.14/) (Yin & Neubig, EMNLP 2022)
- [6] [SyntaxShap: Syntax-aware Explainability Method for Text Generation](https://aclanthology.org/2024.findings-acl.270/) (Amara et al., ACL Findings 2024)
- [7] [Incorporating Attribution Importance for Improving Faithfulness Metrics](aclanthology.org/2023.acl-long.261) (Zhao & Aletras, ACL 2023)

**Questions:**

Minor issues:

- Citations on lines 135, 140, 143, 281 are broken.
- Typo "Fgiure" in lines 186, 196
- Typo "sepcific" in line 248

---

### Note · Authors · 2026-01-13

I have read and agree with the venue's withdrawal policy on behalf of myself and my co-authors.